# Performance of Parallel K-Means Algorithms in Java

Libero Nigro

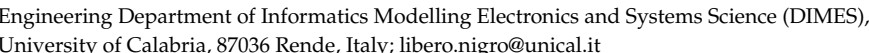

Engineering Department of Informatics Modelling Electronics and Systems Science (DIMES),
University of Calabria, 87036 Rende, Italy; libero.nigro@unical.it

**Abstract:** K-means is a well-known clustering algorithm often used for its simplicity and potential efficiency. Its properties and limitations have been investigated by many works reported in the literature. K-means, though, suffers from computational problems when dealing with large datasets with many dimensions and great number of clusters. Therefore, many authors have proposed and experimented different techniques for the parallel execution of K-means. This paper describes a novel approach to parallel K-means which, today, is based on commodity multicore machines with shared memory. Two reference implementations in Java are developed and their performances are compared. The first one is structured according to a map/reduce schema that leverages the built-in multi-threaded concurrency automatically provided by Java to parallel streams. The second one, allocated on the available cores, exploits the parallel programming model of the Theatre actor system, which is control-based, totally lock-free, and purposely relies on threads as coarse-grain "programming-in-the-large" units. The experimental results confirm that some good execution performance can be achieved through the implicit and intuitive use of Java concurrency in parallel streams. However, better execution performance can be guaranteed by the modular Theatre implementation which proves more adequate for an exploitation of the computational resources.

**Keywords:** parallel algorithms; multi-core machines; K-means clustering; Java; functional parallel streams; actors; message-passing; lightweight parallel programming

## 1. Introduction

The work described in this paper aims to experiment with high-performance computing on nowadays commodity multi/many core machines, which are more and more equipped with significant amount of shared memory and powerful computational units. However, effectively exploiting such computing potential is challenging and requires the adoption of innovative software engineering techniques, hopefully, easy to use by domain specific final users. The paper, in particular, focuses on the support of parallel organizations of the well-known K-Means algorithm [1,2] which is very often used in unsupervised clustering applications such as data mining, pattern recognition, image processing, medical informatics, genoma analysis and so forth. This is due to its simplicity and its linear complexity $O(NKT)$ where $N$ is the number of data points in the dataset, $K$ is the number of assumed clusters and $T$ is the number of iterations needed for convergence. Fundamental properties and limitations of K-Means have been deeply investigated by many theoretical/empirical works reported in the literature. Specific properties, though, like ensuring accurate clustering solutions through sophisticated initialization methods [3], can be difficult to achieve in practice when large datasets are involved. More in general, dealing with a huge number of data records, possibly with many dimensions, create computational problems to K-Means which motivate the use of a parallel organization/execution of the algorithm [4] either on a shared-nothing architecture, that is a distributed multi-computer context, or on a shared-memory multiprocessor. Examples of the former case include message-passing solutions, e.g., based on MPI [5], with a master/slave organization [6], or based on the functional framework MapReduce [7,8]. For the second case, notable are the experiments conducted by using OpenMP [9] or based on the GPU architecture [10].

Distributed solutions for K-Means can handle very large datasets which can be decomposed and mapped on the local memory of the various computing nodes. Shared memory solutions, on the other hand, assume the dataset to be allocated, either monolithically or in subblocks to be separately operated in parallel by multiple computing units.

The original contribution of this paper consists in the development and performance evaluation of two Parallel K-Means solutions in Java which can work with large datasets and can exploit various initialization methods. The first solution is structured according to a map/reduce schema that leverages the built-in multi-threaded concurrency automatically provided by Java to parallel streams [11,12]. The second one depends on the lightweight parallel programming model of the Theatre actor system [13], which is control-based, totally lock-free and purposely reduces the use of threads to be only "programming-in-the-large" units, to be allocated onto the available cores. The experimental results confirm some good execution performance can be achieved through the implicit and intuitive use of Java concurrency in parallel streams. Better execution performance, though, is delivered by the modular Theatre actor-based implementation which seems more adequate for a systematic exploitation of the available computational resources.

The paper is an extended version of the preliminary author's work described in a conference paper (ICICT 2022, Springer LNNS, ISSN: 2367-3370). Differences from the conference paper are (1) the development of a Java parallel stream-based version for K-Means, (2) a complete description of the Theatre actor-based K-Means solution, (3) an implementation of various K-Means initialization methods either stochastic or deterministic [3] which can be exploited by both the parallel stream-based K-Means and the Theatre-based solution, (4) the use of real-world and synthetic large dataset, (5) a detailed experimental work comparing the execution performance which can be achieved by the two Java-based solutions of parallel K-Means.

The remaining of this paper is organized as follows. Section 2 presents some background work on K-Means and about the fundamental issue of the cluster initialization methods. Section 3 proposes two parallel implementations in Java, the first one based on the use of streams and lambda expressions, the second one based on the Theatre actor system. Differences between Theatre actors and classical actors are discussed and the special features introduced by Theatre e.g., for supporting high-performance computing are highlighted. Section 4 describes some experimental results and a performance comparison of the developed solutions, together with an indication of limitations of the approach. Section 5, finally, draws some conclusions and suggests some directions of further work.

## 2. Background on K-Means

In the following, the basic concepts of K-Means are briefly reviewed. A dataset $X = \{x_1, x_2, \ldots, x_N\}$ is considered with $N$ data points (records). Each data point has $D$ coordinates (number of features or dimensions). Data points must be partitioned into $K$ clusters in such a way to ensure similarity among the points of a same cluster and dissimilarity among points belonging to different clusters. Every cluster $C_k$, $1 \leq k \leq K$, is represented by its *centroid* point $c_k$, e.g., its mean or central point. The goal of K-Means is to assign points to clusters in such a way to minimize the *sum of squared distances* (*SSD*) objective function:

$$SSD = \sum_{k=1}^{K} \sum_{\substack{i = 1 \\ x_i \in C_k}}^{n_k} d(x_i, c_k)^2$$

where $n_k$ is the number of points of $X$ assigned to cluster $C_k$ and $d(x_i, c_k)$ denotes the Euclidean distance between $x_i$ and $c_k$. The problem is NP-hard and practically is approximated by heuristic algorithms. The standard method of Lloyd's K-means [14] is articulated in the iterative steps shown in Figure 1.

1. *Initialize* the $K$ centroids $\{c_1, c_2, \ldots, c_k\}$ by some initialization method.
2. *Assign* each data point $x_i \epsilon X$ to cluster $C_k$ according to minimal Euclidean distance $d(x_i, c_k)$.
3. *Update* centroids with the mean point of each cluster, denoted as:

$$c'_k = \frac{1}{n_k} \sum_{h=1}^{n_k} x_h$$

4. *Repeat* from step 2 until a convergence criterion is met.

**Figure 1.** Sequential operation of K-Means.

Several initialization methods for the step 1 in Figure 1, either stochastic or deterministic (see also later in this section), have been considered and studied in the literature, see e.g., [3,15–18], where each one influences the evolution and the accuracy of K-Means. Often, also considering the complexity of some initialization algorithms, the *random* method is adopted which initializes the centroids as random selected points in the dataset, although it can imply K-Means evolves toward a local minima.

The convergence at step 4 can be checked in different ways. In this work, the distance $d(c'_k, c_k)$ is evaluated and when it, for all the centroids, is found to be less than an assumed threshold, convergence is supposed to be reached. A maximum number of iterations can be established to terminate the algorithm in any case.

### 2.1. About the Initialization Methods

Different initialization methods have been proposed and evaluated for K-Means [3,17,18]. Each method is characterized by its computational complexity and its ability to enable an "optimal" clustering solution. Both stochastic and deterministic initialization methods were studied. A deterministic method can generate a (hopefully good) final solution after just one execution run. Generally speaking, initial centroids should be chosen so as to

1. exclude outlier points;
2. be located in regions of high density of local data points;
3. be far from each other so as to avoid, e.g., (since $K$ is fixed) splitting a homogeneous dense cluster, thus degrading the clustering accuracy.

Of course, although it is easy and efficient to also apply to large datasets, the random initialization method addresses none of the abovementioned points. As a consequence, the quality of the generated clustering can be poor. Nonetheless, the reason why the random initialization method is most often used, is tied to the fact that by repeating the execution of K-Means many times, a "good" clustering solution eventually emerges among the various achieved solutions, as one which minimizes the objective function.

Let $D(x_i)$ denote the minimal distance of the point $x_i$ to existing centroids (during the initialization process, the number of centroids $L$ increases, gradually, from 1 to $K$).

The K-Means++ method starts by assigning a random point in the dataset as the first centroid $c_1$ ($L = 1$). Then, a new point $x_j$ can be chosen as the next centroid with probability

$$p(x_j) = \frac{D(x_j)^2}{\sum_{i=1}^{N} D(x_i)^2}$$

After that $L$ is incremented, and the process is continued until $L = K$ centroids are defined.

K-Means++ is an example of an initialization method that tries only to choose centroid points in the dataset that are far from each other (point 3. above). The method is stochastic because, since the use of probabilities, it necessarily has to be repeated multiple times to generate a stable initialization.

A notable example of an initialization method which is robust to outliers and addresses all the above-mentioned tree points, is ROBIN (ROBust INitialization) [3,16], which can

be turned to behave deterministically. ROBIN is based on the Local Outlier Factor (*LOF*), originally introduced by Breunig et al. in [19] which can be assigned to points of the dataset. *LOF* takes into account a density notion around a given point and among close points and permits to filter out outliers during the centroid initialization. The modus operandi of ROBIN is summarised in Figure 2, whereas the formal definition of *LOF* is shown in Figure 3.

---

1. Pick a reference data point $x_r$.

2. Sort data points in decreasing order of their distance from $x_r$.

3. Pick the first data point $x_i$, in sorted order, such that $LOF_{mp}(x_i) \approx 1$, and define $x_i$ as the first centroid $c_1$ and assign $L = 2$.

4. Sort the data points in decreasing order of $D(x_i)$, that is minimal distance from existing centroids.

5. Pick the first data point $x_i'$, in sorted order, such that $LOF_{mp}(x_i') \approx 1$, as the next centroid $c_L$, and put $L++$.

6. If $L \leq K$, repeat from 4.

---

**Figure 2.** The ROBIN initialization method.

---

1. Define density of a data point $x_i$ as:
$$density_{mp}(x_i) = \frac{|N_{mp}(x_i)|}{\sum_{x_i' \in N_{mp}(x_i), i' \neq i} d(x_i, x_i')}$$

2. Establish the *average relative density* $ard_{mp}$ of a data point $x_i$ as:
$$ard_{mp}(x_i) = \frac{density_{mp}(x_i)}{\frac{\sum_{x_i' \in N_{mp}(x_i), i' \neq i} density_{mp}(x_i')}{|N_{mp}(x_i)|}}$$

3. Define the $LOF_{mp}(x_i)$ of data point $x_i$ as:
$$LOF_{mp}(x_i) = \frac{1}{ard_{mp}(x_i)}$$

---

**Figure 3.** Formal definition of the Local Outlier Factor (*LOF*) of a data point.

The operation of the ROBIN algorithm in Figure 2 and the *LOF* definition in Figure 3 depend on the parameter $mp$, that is the *minimum number of points* in the "nearest" neighborhood of a point $x_i$. Such a parameter is directly related to the *k-distance-neighborhood* introduced in [19]. Upon $mp$ is based the composition of the neighborhood set $N_{mp}(x_i)$ which can be established when a new nearest point to $x_i$ is met such that at least $mp$ points were previously already found to be closer to $x_i$. Hence, the cardinality of the neighborhood set $|N_{mp}(x_i)|$ is necessarily expected to be $\geq mp$. As it emerges from Figure 3, the *LOF* value takes into account not only the density of a point $x_i$, but also the *reciprocal* density of points *near* to $x_i$. In fact, for not outlier points, the density of $x_i$ and that of neighbor points should be almost equal. Therefore, a point is a candidate for the next centroid, provided its *LOF* is *close* to 1. Values of $LOF \gg 1$ denote outliers. As in [3], a tolerance $E$ (e.g., 0.05) can be defined and a *LOF* value is assumed to be that of a *not* outlier when: $1 - E < LOF < 1 + E$.

Complexity of the ROBIN initialization method is $O(NlogN)$ as it is dominated by repeated sorting of the dataset at each new centroid definition. Such a computational cost can be a problem in very large datasets. Anyway, the ROBIN method can be preferable to other initialization methods like the Kaufman method [3], which has a cost of $O(N^2)$ being necessary to re-compute the distance between every pair of points.

As a final remark, the ROBIN method is stochastic when the reference point at the point 1 in Figure 2 is chosen randomly in the dataset. It becomes deterministic, as in the original work [16], when the reference point is assumed to be the origin.

### 3. Parallel K-Means in Java

In the basic K-Means algorithm shown in Figure 1, there is a built-in parallelism in both the *Assign* and the *Update* phases, which can be extracted to speed-up the computation. In other terms, both the calculation of the minimal distances of points $x_i$ to current centroids, and then the definition of new centroids as the mean points of current clusters, can be carried in parallel.

*3.1. Supporting K-Means by Streams*

A first solution to host parallel K-Means in Java is based on the use of streams, lambda expressions and a functional programming style [12], which were introduced since the Java 8 version. Streams are *views* (not copies) of collections (e.g., lists) of objects, which make it possible to express a fluent style of operations (method calls). Each operation works on a stream, transforms each object according to a lambda expression, and returns a new stream, ready to be handled by a new operation and so forth. In a fluent code segment, only the execution of the terminal operation actually triggers the execution of the intermediate operations.

Figure 4 depicts the main part of the K-Means solution based on streams which can execute either serially or in parallel. The two modes are controlled by a global Boolean parameter *PARALLEL* which can be *false* or *true*. The algorithm first loads the dataset and the initial centroids onto two global arrays of data points (of a class *DataPoint*), respectively *dataset* and *centroids* of length $N$ and $K$, from which corresponding (possibly parallel) streams are built.

A Map/Reduce schema was actually coded where the Map stage corresponds to the *Assign* step of Figure 1, and the Reduce stage realizes the *Update* step of Figure 1.

The *DataPoint* class describes a point in $R^D$, that is with $D$ coordinates, and exposes methods for calculating the Euclidean distance between two points, adding points (by adding the corresponding coordinates and counting the number of additions), calculating the mean of a point summation and so forth. *DataPoint* also admits a field *CID* (Cluster Identifier) which stores the *ID* (index of the *centroids* array) of the cluster the point belongs to. The *CID* field is handled by the *getCID()*/*setCID()* methods. The *map()* method on *p_stream* receives a lambda expression which accepts a *DataPoint* $p$ and returns $p$ mapped to the cluster centroid closer to $p$. Since *map()* is an intermediate operation, a final fictitious *forEach()* operation is added which activates the execution of *map()*.

When *p_stream* is completely processed, a *newCentroids* array is reset (each new centroid has 0 as coordinates) so as to generate on it the updated version of centroids. Purposely, the *CID* field of every new centroid is set to itself.

The *c_stream* is actually extracted from the *newCentroids* array. The *map()* method on *c_stream* receives a lambda expression which accepts a new centroid point $c$ and adds to $c$ all the points of the dataset whose *CID* coincides with the *CID* of $c$. Following the summation of the points belonging to the same new centroid $c$, the *mean()* method is invoked on $c$ to calculate the mean point of the resultant cluster.

The iteration counter *it* gets incremented at the end of each K-Means iteration. The *termination()* method returns *true* when the convergence was obtained or the maximum number of iterations $T$ was reached. In any case, *termination()* ends by copying the contents of *newCentroids* onto the *centroids* array. Convergence is sensed when all the distances among new centroids and current centroids fall under the threshold $THR$.

```
load_dataset();
load_centroids();
long start=System.currentTimeMillis();
do{
    //Map stage-assign data points to clusters
    Stream<DataPoint> p_stream=Stream.of (dataset);
    if (PARALLEL) p_stream=p_stream.parallel();
    p_stream
        .map (p -> {
            double md=Double.MAX_VALUE;
            for (int k=0; k<K; ++k) {
                double d=p.distance(centroids[k]);
                if (d<md) { md=d; p.setCID(k); }
            }
            return p; })
        .forEach (p->{});
    //prepare newCentroids
    for(int i=0; i<K; ++i) {
        newCentroids[i].reset();
        newCentroids[i].setCID (i);
    }
    //Reduce stage-define newCentroids
    Stream<DataPoint> c_stream=Stream.of (newCentroids);
    if (PARALLEL) c_stream=c_stream.parallel();
    c_stream
        .map (c -> {
            for (int i=0; i<N; ++i) {
                if (dataset[i].getCID()==c.getCID()) c.add (dataset[i]);
            }
            c.mean();
            return c;})
        .forEach(c->{});

    it++;
}while (!termination());
long end=System.currentTimeMillis();
…/*output operations*/
```

**Figure 4.** A map/reduce schema for K-Means based on Java streams.

It is worthy of note that all the globals: the parameters *N*, *K*, *D*, *T*, *THR*, *E*, *it*, *MP*, *PARALLEL* . . . , the *dataset*, *centroids*/*newCentroids* data point arrays, the implemented centroid initialization methods (all the ones, either stochastic or deterministic, discussed in [3]), some auxiliary methods for calculating the *SSD* cost of a clustering solution or evaluating its quality by, e.g., the *Silhouette* index [3] and so forth, are introduced as static entities of a *G* class, which are statically imported in applicative classes for a direct access to its entities.

The Java code in Figure 4 automatically adapts itself to a multi-threaded execution, when the *p_stream* and *c_stream* are turned to a parallel version. In this case, behind the scene, Java uses multiple threads of a thread-pool to accomplish in parallel, on selected data segments, the underlying operations required by the Map and Reduce phases.

A basic feature of the Java code is its independence from the order of processed data points and the absence of side-effects which can cause interference problems among threads

when accessing to shared data. During the Map stage, the lambda expression operates on distinct data points and affects their $CID$ without any race condition. Similarly, multiple threads act on distinct new centroid points during the Reduce phase, and safely accumulate data points of a same cluster.

A specific parameter in the $G$ class is $INIT\_METHOD$ which takes values in an enumeration with the constants $RANDOM$, $MAXIMIN$, $KMEANSPP$, $KAUFMAN$, $ROBIN$, $DKMEANSPP$, to select a corresponding centroid initialization method [3]. Other initialization methods can be added as well. The ROBIN method, which can be configured to operate stochastically or deterministically, relies concretely on heap-sorting (through a *PriorityQueue* collection) the dataset in descending order of the minimal distances from the existing centroids and initially from the reference point. In addition, the nearest $MP$-neighborhood of a data point $x_i$, considered in sorted order, is determined by moving around $x_i$, and registering in a sorted data structure the distances of nearest points to $x_i$. Movement is stopped as soon as a point is encountered whose distance from $x_i$ is greater than that of any already met nearest point, and the cardinality of previous visited nearest points is found to be $\geq MP$.

The $G$ class is extended by a further parameter $P$ which specifies the degree of parallelism, that is the number of the underlying cores (threads) exploitable in the execution experiments.

### 3.2. Actor-Based K-Means Using Theatre

Another solution for serial/parallel K-Means was achieved on top of the Theatre actor system. Theatre is both a formal modelling language [20] and an implementation framework in Java. It addresses modelling, analysis and synthesis of parallel/distributed time-dependent systems like cyber-physical systems with strict timing constraints [20,21].

A key difference from the classical actor computational model [22,23] is the adoption of a (transparent) *reflective control-based layer* which can reason on a time notion (real-time or simulated-time) or on no-time (for concurrent/parallel systems), and regulates the ultimate delivery order of the asynchronously exchanged messages among actors, which in [22] is truly non-deterministic.

Theatre is characterized by its lightweight and totally lock-free architecture. The design purposely minimizes the use of threads as only "programming-in-the-large" units, to be allocated on the available cores. The goal is to favor time predictability, as well as the development of high-performance parallel applications [13].

### 3.2.1. The Parallel Programming Model of Theatre

A system consists of a federation of computing nodes (*theatres*) which can be allocated to distinct cores of a multi-core machine. A theatre node is mapped onto a separate thread and is organized into three layers (see also [13]): (1) a *transport-layer,* which is used for sending/receiving inter-theatre messages); (2) a *control-layer* which provides the basic services for scheduling/dispatching of messages; (3) an *application-layer* which is a collection of local business actors.

Both intra-theatre and inter-theatre communications (message exchanges) are enabled. In addition, actors can be moved from a theatre to another, for configuration/load-balancing issues.

Within a same theatre, actors execute according to a *cooperative concurrency* model, ensured by *message interleaving*, that is dispatching and executing one message at a time. Actors are without an internal thread. Therefore, they are at rest until a message arrives. A message is *the* unit of scheduling and dispatching. Message processing is *atomic* and cannot be pre-empted nor suspended. In other terms, messages of any actor, naturally execute in mutual exclusion.

Actor executions (i.e., message processing) into distinct theatres can effectively occur in time-overlapping, that is truly in *parallel*. Since the lock-free schema adopted by Theatre, shared data among actors assigned to distinct theatres/cores, should be avoided to prevent

data inconsistencies. Sharing data, though, among the actors of a same theatre, is absolutely safe due to the adopted cooperative concurrency schema.

A *time server* component, attached to a selected theatre, gets transparently contacted (through hidden *control messages* [13]) by the various control layers of the computing nodes, so as to regularly update the global time notion of a Theatre system. In a pure-concurrent system, a "time server" can be exploited to detect the termination condition of the parallel application which occurs when all the control-layers have no pending messages to process and there are no in-transit messages across theatres.

### 3.2.2. Mapping Parallel K-Means on Theatre

The simplified UML diagram of Figure 5 shows some developed Java classes for supporting Parallel (but also serial) K-Means using Theatre.

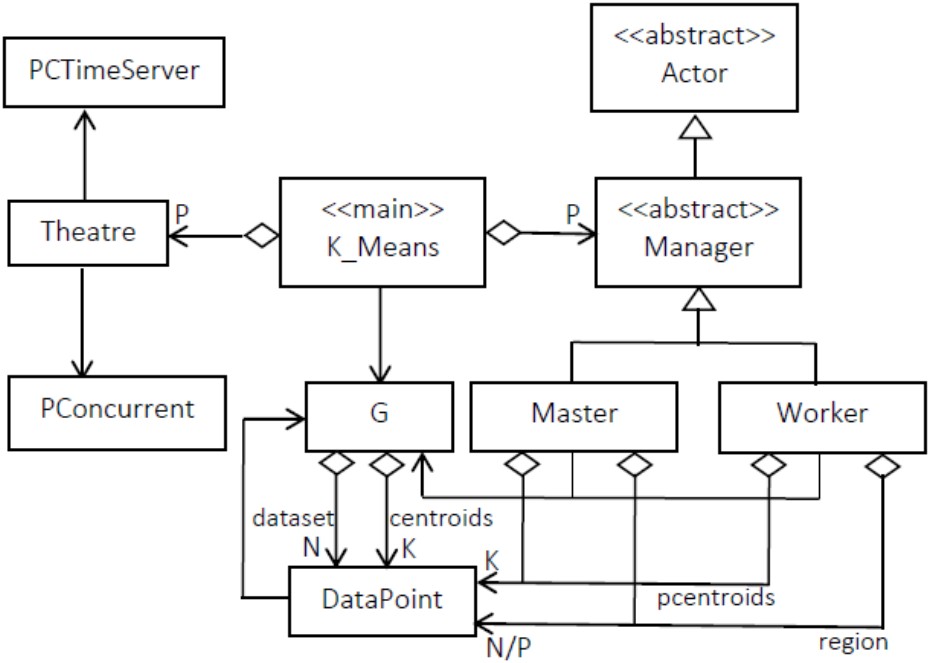

**Figure 5.** Supporting classes for Parallel K-Means based on Theatre.

Basic classes of the Theatre infrastructure include *Actor*, *Theatre*, *PConcurrent* and *PCTimeServer* (see [13] for more details). A programmer-defined actor class must inherit, directly or indirectly, from the *Actor* abstract base class, which exposes the basic services of (unblocking) message *send*, and the *move* operation. The *send* operation rests on Java *reflection* for associating a message name (string) to a message server (method), and for passing arguments at the message delivery time, regulated by the control-layer. Theatres of a system are identified by unique identifiers from 0 to $P - 1$.

$$void\ send\ (String\ message\_name,\ Object\dots args);$$

$$void\ move\ (int\ theatre\_id);$$

A sent message gets (transparently) scheduled on the underlying control-layer. The *move* operation transfers an actor to a given (target) theatre, so that its relevant messages are ultimately handled by the target theatre control-layer. *PConcurrent* and *PCTimeServer* represent respectively the concurrent control-layer and the time server used in a parallel untimed application. Timed versions of these classes are described in [13]. *PConcurrent* ensures messages are delivered in the sending order. By convention, the time server is associated with the theatre 0 which often plays the role of the master theatre.

The actual actor programming style can be checked in Figure 7. An actor admits a local hidden data status which also specifies the *acquaintances*, that is the Java references to known actors (including itself for proactivity) to which this actor can send messages. Local data can only be modified by processing messages. The message interface is specified by the exposed *message server* methods, provided by the annotation *@Msgsrv*, whose names *are* the message names. Message servers are scheduled by the *send* operation and finally dispatched according to the discipline enforced by the control-layer. An actor can also introduce normal helper methods (without the annotation *@Msgsrv*), to facilitate its code structuring.

As indicated in Figure 5, and also shown by the message exchanges abstracted in Figure 6, the Theatre based K-Means adopts a *master/worker* organization (see later in this section for further details).

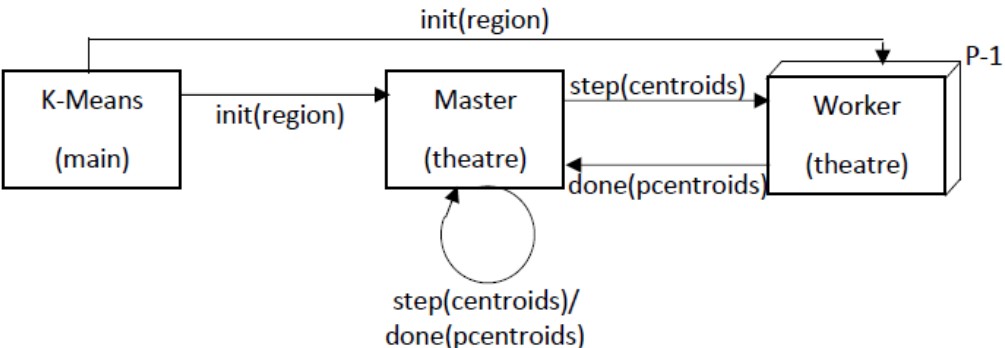

**Figure 6.** Master/worker organization of Theatre-based K-means.

The dataset is equally split in $P$ regions (subblocks) ($P$ is the number of used theatres/cores), each region being (almost) of size $N/P$.

The *main*() method of the *K_Means* class (see Figure 7) instantiates $P$ theatres, each one equipped with an instance of *PConcurrent* and one of *PCTimeServer*. The *run*() method of the master theatre is redefined for completing the parallel configuration. $P$ Manager actors are created, one of the type *Master* and $P-1$ of the *Worker* type. Such actors are moved to separate theatres. For brevity, Figure 8 shows only the *Worker* actor class. Each manager understands the basic *step*() message (see the *step*(...) message server in Figure 8), introduced by the base abstract *Manager* class. As a rule, every programmer-defined actor class is initialized by an explicit *init*(...) message (see also Figure 6), whose parameter list is tuned to the specific task/role assigned to the actor. The *init*(...) of *Master* receives the master ID, the assigned subblock of the dataset, the initial values of the centroids and the array of all the manager references (acquaintances). The *init*(...) of *Worker*, instead, receives (see Figure 8) its unique ID, the assigned subblock of the dataset, and the reference to the *Master* manager.

Configuration ends by the master theatre which activates all the theatres and finally launches, on itself, the execution of the control-layer by invoking the *controller*() method.

The master actor receives a *done*() message from a worker when the latter finishes the operations (Assign and Update sub-phases) of the current step (see Figure 7). The *done*() message carries as an argument the *partial* new centroids calculated according to the *viewpoint* of the worker. When all the *done*() messages of the current step are received, the master combines the partial centroids provided by the workers, and determines the resultant *new* centroids. A helper method of the *Master* actor checks the termination condition which occurs (convergence) when, for each centroid, the Euclidean distance between the new centroid and the current one is found to be less than or equal to the threshold $THR$ (see Table 1) or the maximum number of iterations $T$ was reached. It is important to note that each partial centroid point returned by a worker contains the cumulative $D$ features (coordinates) of the local points handled by the worker, together

with the number of points belonging to the same cluster. This way, the master actor can correctly assemble all the received partial centroids.

```
public class KMeans{
    public static void main(String[] args) throws IOException{
        for (int t=1; t<P; ++t)
            new Theatre(t,P,new PTransportLayer(), new PConcurrent());
        new Theatre(0, P, new PTransportLayer(new PCTimeServer()), new PConcurrent()) {
            public void run() {
                load_dataset(); load_centroids();
                long start=System.currentTimeMillis();
                Manager[] m=new Manager[P];//creates managers one per theatre
                int strip=N/P, residual=N%P;//number of subblocks
                int inf=0, sup=strip-1;//lower and upper bounds of current subblock
                for (int t=0; t<P; ++t) {
                    if ( residual>0) { sup++; residual--; }//distribution of residual data points
                    DataPoint[] subblock=dataset_subblock(inf, sup);
                    if (t==0) {
                        m[t]=new Master();
                        m[t].send("init", t, subblock, centroids, m);
                    }
                    else {
                        m[t]=new Worker();
                        m[t].send("init", t, subblock, m [0]);
                    }
                    m[t].move(t);
                    inf=sup+1; sup=sup+strip;//bounds update for the next subblock
                }
                for (int t=0; t<P; ++t) Theatre.getTheatre(t).activate();//theatre activation
                Theatre.getTheatre(0).getControlMachine().controller();//start of message loop
                long elapsed=System.currentTimeMillis()-start;
                System.out.println("PET="+elapsed);
                …/*output operations*/
            }//run
        };//new Theatre(0,...)
    }//main
}//KMeans
```

**Figure 7.** Configuration program for parallel K-Means based on Theatre.

The master actor increments the iteration counter *it* at the beginning of each iteration. Each iteration is started by the master which broadcasts a $step()$ message to each worker and to itself. The $step()$ message carries as a parameter, the current values of the centroids. At the end of K-Means, the master actor copies the emerged centroids upon the *centroids* variable of the *G* class so that it can be eventually output by the main program. Each centroid point carries the number of dataset points which compose the corresponding cluster. In addition, each dataset point holds the final cluster id (*CID*) of the belonging cluster.

```
public class Worker extends Manager{
    private DataPoint[] region, partial;
    private int ID;
    private Master m;//acquaintance
    @Msgsrv
    public void init (Integer ID, DataPoint[] region, Master m) {
        this.ID = ID;
        this.region=region;//a sub-block of the whole dataset
        this.m = m;
        partial = new DataPoint[K];//used to build partial new centroids
        for (int i = 0; i < K; ++i)          partial[i] = new DataPoint();
    }//init
    @Msgsrv
    public void step (Integer id, DataPoint[] centroids) {
        //Map stage: allocate local data points to clusters
        for (int i = 0; i < region.length; ++i) {
            double minD = Double.MAX_VALUE,d = 0;
            for (int j = 0; j < K; ++j) {
                d = region[i].distance(centroids[j]);
                if (d < minD) { minD=d; region[i].setCID(j); }
            }
        }
        //Reduce stage: updates partial centroids
        for (int i = 0; i < K; ++i) partial[i].reset();
        for (int i = 0; i < region.length; ++i) {
            int cid = region[i].getCID();
            partial[cid].add (region[i]);
        }
        m.send ("done", ID, partial);
    }//step
}//Worker
```

**Figure 8.** The Worker actor class.

**Table 1.** Values of some configuration parameters.

| Parameter | Value | Meaning |
|---|---|---|
| P | 16 | Number of used underlying threads |
| N | 2,458,285 | Size of the Census1990 data set |
| D | 68 | Dimensions of the dataset records |
| K | e.g., 80 | Number of assumed centroids |
| T | 1000 | Maximum number of iterations |
| THR | $1 \times 10^{-10}$ | Threshold for assessing convergence |
| INIT_METHOD | ROBIN | The centroid initialization method |
| E | 0.05 | Tolerance in the Local Outlier Factor (LOF) detection (ROBIN) |
| MP | e.g., 15 | Size of the MP-neighborhood of data points (ROBIN) |

The actor-based parallel K-Means can be easily adapted to work with one single theatre/core, thus achieving the *standalone* K-Means program, useful for performance comparisons. Except for the synchronization messages exchanged at each step of the algorithm (broadcast of *step()* messages by the master to workers, followed by the reply

*done*() messages sent by workers to the master), the parallel and the standalone K-Means programs execute *exactly* the same number of operations.

## 4. Experimental Results

The Java-based K-Means (KM) algorithms were thoroughly checked on different datasets, real-world or synthetic. It was ensured that, starting from the same initial centroids, the four versions of the developed solutions, namely the Serial Stream KM (*SSKM*), the Parallel Stream KM (*PSKM*), the Standalone Theatre KM (*STKM*) and the Parallel Theatre KM (*PTKM*), always and exactly generate the same final centroids and with the same number of convergence iterations. For demonstration purposes, the following considers the US Census Data 1990, downloaded from the UCI Machine Learning Repository [24]. The dataset (see also Table 1) contains $N = 2458285$ data records each one with $D = 68$ numerical categorial attributes. Different clustering solutions were studied for the chosen dataset (see the $K$ number of assumed centroids in the Table 1), under the deterministic version of the ROBIN initialization method (see Section 2.1). Performance measurements are documented in the author's preliminary work (ICICT 2022, Springer LNNS, ISSN: 2367-3370) using some synthetic datasets when the number N of data points increases and the random initialization method (see Section 2) is adopted.

All the execution experiments were carried out on a Win10 Pro, Dell XPS 8940, Intel i7-10700 (8 physical cores which with the hyperthreading give support to 16 threads), CPU@2.90 GHz, 32GB Ram, Java 17 with the default JVM configuration parameters (e.g., 4GB for the maximum exploitable heap memory size). Table 1 collects the values of some basic parameters. It is useful to note that the value $P = 16$ is implicitly and automatically exploited by Java when working with *PSKM*. Performance comparison was also explicitly set for the case of PTKM.

Some preliminary runs were devoted to studying the K value of K-means, which can minimize the SSD cost (sum of squared distances objective function—see Section 2). Figure 9 depicts the so-called "Elbow" curve, i.e., the SSD vs.K, which is often used to understand the values of K to adopt, and Figure 10 shows the number of emerged iterations required for convergence, i.e., it vs.K. The execution performance was then investigated by varying the K value from 10 to 120, as shown in Table 2.

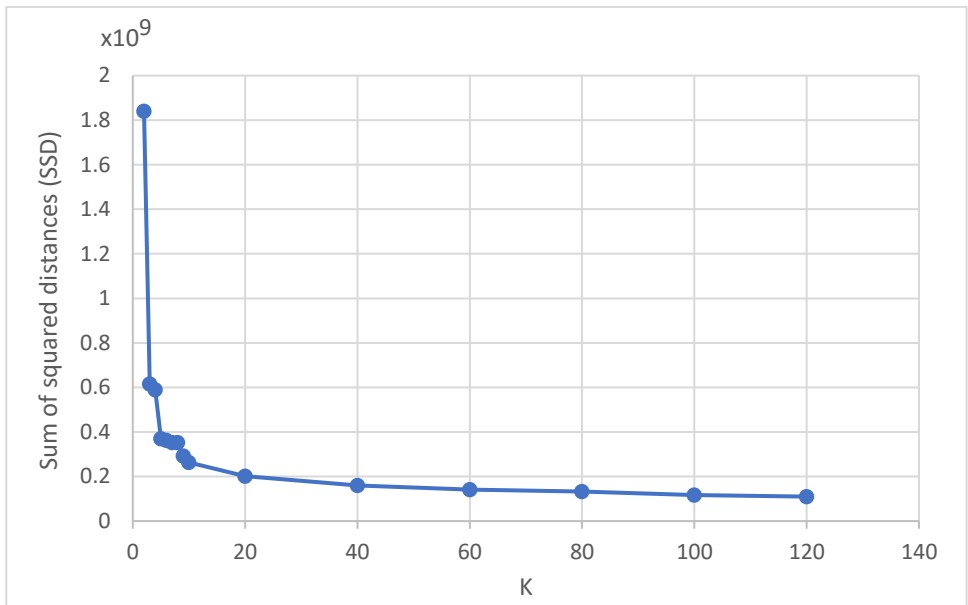

**Figure 9.** The "Elbow" curve, i.e., the *SSD* cost vs. *K*.

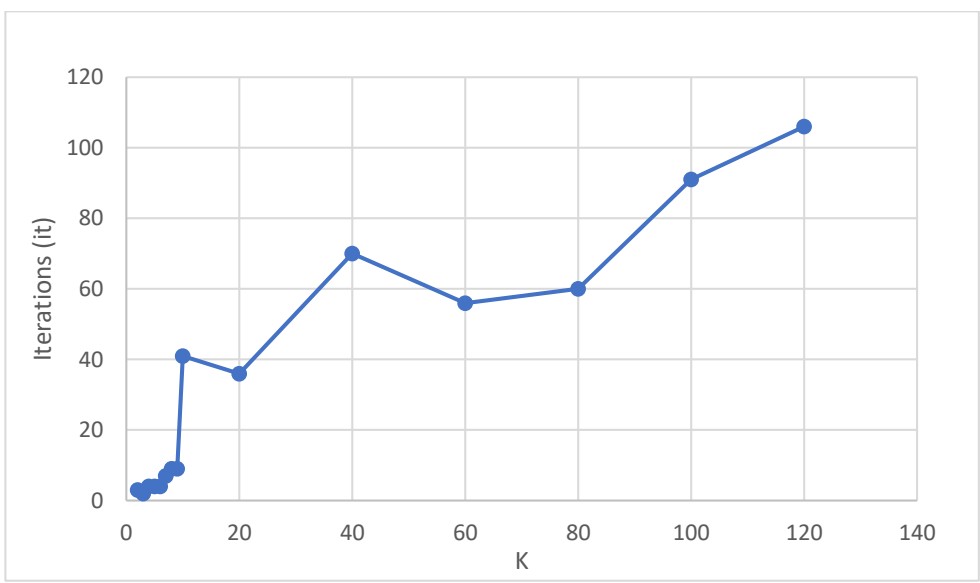

**Figure 10.** The number of iterations (*it*) for convergence vs. *K*.

**Table 2.** Serial/parallel elapsed times for Stream-based and Theatre-based K-Means vs. K (P = 16).

| *K* | $SET_{SSKM}$ (ms) | $PET_{PSKM}$ (ms) | $Speedup_S$ | $SET_{STKM}$ (ms) | $PET_{PTKM}$ (ms) | $Speedup_T$ |
|---|---|---|---|---|---|---|
| 10 | 70,566 | 9233 | 7.64 | 59,671 | 7620 | 7.83 |
| 20 | 114,743 | 12,061 | 9.51 | 96,891 | 11,221 | 8.63 |
| 40 | 428,389 | 44,877 | 9.55 | 357,731 | 38,535 | 9.28 |
| 60 | 510,496 | 53,798 | 9.49 | 420,934 | 45,217 | 9.31 |
| 80 | 769,109 | 68,158 | 11.28 | 597,489 | 63,497 | 9.41 |
| 100 | 1,407,401 | 145,346 | 9.68 | 1,117,578 | 111,956 | 9.98 |
| 120 | 1,920,530 | 199,335 | 9.63 | 1,552,306 | 146,059 | 10.63 |

For the purpose of the performance comparison work, documented in Table 2, the deterministic pre-selection of the initial centroids of the ROBIN method was preferred. In particular, the initial centroids, for each value of *K*, were saved on disk and reused (by re-loading) without regenerating them when executing the various versions of the K-Means algorithm. As in [19], it emerged that the determination of the *LOF* factor for outlier data points, is not much sensitive to the value of *MP* (see Table 1). The value $MP = 10$ was used.

Table 2 reports the measured serial elapsed time (*SET*) and the parallel elapsed time (*PET*) for the various versions of K-Means and for the chosen values of *K* which minimize the *SSD* cost (see Figure 9). Each measure is the average of five runs which, although the deterministic evolution of K-Means due to the adopted ROBIN initialization method, are required to smooth out uncertainties of the underlying Operating System.

As one can see from Table 2, the parallel Stream-based solution (*PSKM*) is capable of delivering a good execution performance, with the burden of choosing the parallelism-degree, splitting the dataset in segments and processing them concurrently by separate threads is completely left to the Java runtime system. The emerged speedup for the Stream-based solutions, that is the ratio between the $SET_{SSKM}$ and the $PET_{PSKM}$, is shown in the column $Speedup_S$ and it reaches a value about 11 in the case $K = 80$. The corresponding observed times when the Theatre-based solutions are used, are indicated in the second part of Table 2, confirming a speedup of 10.63 in the scenario with $K = 120$.

Although the observed values of $Speedup_S$ and $Speedup_T$ are similar, from the Table 2 it emerges that the Theatre actor-based solutions perform better than the corresponding

Stream-based solutions, both in the serial and the parallel cases, which seems to indicate Theatre is more able to exploit the underlying computational resources. This is also confirmed by the measured $Speedup_T$ which scales more regularly with the increase of the number $K$ of clusters.

The registered speedup values are in many cases examples of *super-speedups*, when one considers that the number of physical cores of the used machine is 8. Such super-speedup closely mirrors the number of physical and hardware emulated cores, and the important contribution of the L2 cache of cores, where segments of the dataset are pre-loaded and re-used thus avoiding, very often, the need to accessing data in the internal memory. Such a phenomenon is well documented in the literature, see e.g., [13,25,26] and it was formally predicted and practically observed in the parallel processing of large numerical arrays and matrices [25].

As a final remark, the developed stream-based versions of K-Means as well as the standalone Theatre-based version, have the limitation of allocating the whole dataset in a native array collection, which the parallel stream version can then split into consecutive segments and process them by separate threads. Of course the approach can be difficult to apply to some large datasets, although the memory size of current commodity multi-core machines is noticeable. On the other hand, the Theatre actor-based parallel version of K-Means is capable of feeding the master/worker actors with consecutive pieces of the dataset, directly read from the source file. In any case, the handling of very large datasets could be possible by using a multi-computer solution, based on distributed Theatre, which can partition the work and the data among the machines of a networked system, but this is further work to do.

## 5. Conclusions

This paper proposes two original parallel implementations in Java of the K-Means algorithm [1,2,14] which can deliver good execution performance when handling large datasets on nowadays commodity multi/many-core machines with shared memory. The first implementation depends on the use of Java streams and lambda expressions [11,12]. The second one is based on an efficient actor-based system named Theatre [13] which exposes an easy to use light-weight parallel programming model, totally lock-free. Although the solution based on parallel streams is notable, being simple and intuitive to program and understand, the Theatre-based solution seems more apt to furnish a higher execution performance in the practical case.

It is planned to prosecute the described work as follows:

- Systematically exploiting the developed Java approach to prove, empirically, the properties of K-Means, using selected and challenging datasets [3,15].
- Extending and experimenting with the set of supported robust to outliers initialization methods for K-Means [3,27–31].
- Adapting the approach for studying variations of the K-Means clustering [3].
- Porting the parallel Theatre-based version of K-Means to a distributed context, so as to cope with very large datasets.

**Funding:** This research received no external funding.

**Institutional Review Board Statement:** Not applicable.

**Informed Consent Statement:** Not applicable.

**Data Availability Statement:** Data is contained within the article.

**Conflicts of Interest:** The authors declare no conflict of interest.

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
