# Peer review of "Performance of Parallel K-Means Algorithms in Java"

_algorithms, doi:10.3390/a15040117_

Round 1

Reviewer 1 Report

REMARKS ON CONTENTS OF THE ARTICLE

  1. There is no “Related work” part in the article. I believe this is a “must-have” section for every serious scientific article. Such a section should contain more references than in Introduction and describe them in more detail (not just mentioning/naming than in the Introduction). In the Introduction, the author omits recent references that are close to the article’s topic. GPU and CUDA parallel implementations of k-means are missed (e.g., S. Cuomo, et al.: A GPU-accelerated parallel K-means algorithm. Comput. Electr. Eng. 75: 262-274 2019, https://doi.org/10.1016/j.compeleceng.2017.12.002). Next, Java implementations that could serve as competitors to the author’s approach are not mentioned (e.g., V. Sudarsan, et. al: Building a distributed K-Means model for Weka using remote method invocation (RMI) feature of Java. Concurr. Comput. Pract. Exp. 31(14) 2019, https://doi.org/10.1002/cpe.5313).
  2. In the experiments, the authors do not compare their solution with any rival approach (on the same platform and over the same dataset). I believe this is a “must-have” aspect of the experimental evaluation to understand the place of our approach among the analogs. Is the author’s approach better (in terms of performance, memory, quality of clustering, etc.) than rivals (that should be clearly described in the Related work section), with respect to what kind of data, etc?
  3. In the experiments, the author shows only the absolute speedup of the proposed solution (i.e. the ratio between the running time on a single core and the same one for 16 threads running on 8 cores, see Tab. 2). However, this is not helpful to understand the scalability of the author’s approach. At first, the parallel speedup (or simply speedup later) is a much more important characteristic of a parallel algorithm since it shows the scalability of the algorithm. A speedup is defined as the ratio of the algorithm’s running time for one thread to the running time on p threads for different values of p (while the problem size remains the same and the number of clusters is fixed). From this point of view, it can be recommended to evaluate the proposed approach on the Intel Xeon Gold (Cascade Lake) platform that provides 18 cores with 2-factor hyper-threading. Will the author’s solution show linear or close-to-linear speedup? Next, it can be recommended to address how well the algorithm scales along with the problem size, and evaluate the algorithm’s scaled speedup. It is defined as the speedup obtained when the problem size is increased linearly with the number of the threads (see A. Grama, et al.: Introduction to Parallel Computing. 2nd ed. Boston, MA, USA: Addison-Wesley, 2003). Again, will the author’s solution show linear or close-to-linear scaled speedup?

REMARKS ON FORMATTING OF THE ARTICLE

  1. English would be improved. E.g.,
  • in line 36: need a comma after “hopefully”
  • in line 36: “domain specific” to be changed to “domain-specific”
  • in line 371: “ESPERIMENTAL RESULTS” to be changed to “EXPERIMENTAL RESULTS”
  • in lines 24, 67, 451: “realization” to be changed to “implementation”.
  1. In Fig. 4, 6, 7, the author employs Java listings that look cumbersome and unclear. A pseudo-code style is more appropriate for such a mathematical case. In addition, keywords given in bold can help to understand the algorithms.

Reviewer 2 Report

This work assesses the performance of two Java implementations of the K-means clustering algorithms. While the theoretical contribution is almost absent, this study main part are the experimental results of the performance of the k-means implementations.

The paper is well structured, the references are relevant for this field, and the methods are adequately presented.

However, some clarifications/corrections should be added before publication:

  • The English language should be revised (see for example the title of section 4: ESPERIMENTAL RESULTS)
  • Some clarifications of the results:
    • Lines 430-433: is it not possible that the speedup greater than 8 (the number of physical cores) could come from the usage of the logical cores and not only from L2 cache?
    • Do you have any control on the number of threads in both implementations?
    • The results show only the execution time for different values of k. Was an analysis performed based on the number of data points?

Reviewer 3 Report

The author claims that the paper is an extended version of previously published work. One of the points states that the paper contains a complete description of the Theatre actor-based solution. However, reading sections 3.2.1 and 3.2.2 show a lack of some fundamental definitions, e.g. "precedence constraint rule" and "hidden control messages" are only mentioned, and the details are presented in other work. That is a minor flaw. The bigger one is, for example, the move operation that is not described at all (only definition described). There is more flaws.

Code listings have to be described in detail. There are many methods, parameters, constants that are only on the listings or only mentioned in the manuscript text but without description.

I also do not see "an implementation of various K-Means initialization methods either stochastic or deterministic" in the manuscript.

The author claims that the paper contains an implementation of K-Means based on Java streams. However, there is no real usage of Java streams. The map method body should be entirely moved to the forEach method, as the output of the map is not used at all. Then the Java streams are only reduced to invoking the processing sequentially or in parallel using the default Fork-Join pool created by Java Virtual Machine (JVM) itself. I do not think it can be called "implementation based on Java streams".

The description of experimental ("esperimental" in manuscript) results lacks information on JVM set up, how the author warmed up the JVM, how many repetitions of experiments were performed and what kind of statistical function was used to collect the presented results. That is important as benchmarking applications using JVM is not trivial and depends on many factors.
I feel that the results of "Java streams" and "Theatre" based implementation are not fair, as on the listings (Fig. 4 and Fig. 6), the measured time in "Java streams" counts almost everything, but Theatre lacks measuring required initial steps.

Round 2

Reviewer 1 Report

The authors have carefully taken into account all the comments, so I consider it possible to accept the manuscript in its current form with the following small formatting improvements: 1) in Figure 9, leave numbers up to 10 on the ordinate axis, and add a multiplier of 10^9 to the axis caption 2) present Table 2 in the form of two graphs (Running time depending on K for two cases, indicating speedup at each point) to improve clarity.

Reviewer 3 Report

The updated version is much better, especially with the cover letter that aims my concerns.

However, I think that the in-depth explanation of the statement that the solution using “Java Streams” have to be added to the manuscript, as potential readers of the paper will have a text of the manuscript available only.

Moreover, the presented code fragments are not well written to use with Java Streams. For example, the c_stream, which purpose is generating centroids with dataset points, could be written in reverse order, so process datapoints and group them by CID, then make centroids on them and calculate mean of them. Currently, the list of data points is traversed as many times as a number of centroids (K). The solution for Theatre presented in Fig.8 traverses region (datapoints) only once.

One more thing that I have missed in the previous review is the “High-performance computing” (HPC) keyword, as other aspects look more substantial. I don’t see anything special about HPC in the paper – it is just “normal” computing, especially as only one computing node is used. I recommend removing that keyword from the list. Only the introduction mentions experimentation with high-performance computing on multi/many-core machines, but many cores do not mean HPC. I think that also should be corrected.
